# Wnt signaling modulates the response to DNA damage in the *Drosophila* wing imaginal disc by regulating the EGFR pathway

**Ben Ewen-Campen**[1]*, **Norbert Perrimon**[1,2]*

**1** Department of Genetics, Blavatnik Institute, Harvard Medical School, Boston, Massachusetts, United States of America, **2** Howard Hughes Medical Institute, Boston, Massachusetts, United States of America

* bewencampen@genetics.med.harvard.edu (BE-C); perrimon@genetics.med.harvard.edu (NP)

**Data Availability Statement:** All relevant data are within the paper and its Supporting Information files.

## Abstract

Despite the deep conservation of the DNA damage response (DDR) pathway, cells in different contexts vary widely in their susceptibility to DNA damage and their propensity to undergo apoptosis as a result of genomic lesions. One of the cell signaling pathways implicated in modulating the DDR is the highly conserved Wnt pathway, which is known to promote resistance to DNA damage caused by ionizing radiation in a variety of human cancers. However, the mechanisms linking Wnt signal transduction to the DDR remain unclear. Here, we use a genetically encoded system in *Drosophila* to reliably induce consistent levels of DNA damage in vivo, and demonstrate that canonical Wnt signaling in the wing imaginal disc buffers cells against apoptosis in the face of DNA double-strand breaks. We show that Wg, the primary Wnt ligand in *Drosophila*, activates epidermal growth factor receptor (EGFR) signaling via the ligand-processing protease Rhomboid, which, in turn, modulates the DDR in a *Chk2*-, *p53*-, and *E2F1*-dependent manner. These studies provide mechanistic insight into the modulation of the DDR by the Wnt and EGFR pathways in vivo in a highly proliferative tissue. Furthermore, they reveal how the growth and patterning functions of Wnt signaling are coupled with prosurvival, antiapoptotic activities, thereby facilitating developmental robustness in the face of genomic damage.

## Introduction

In response to DNA damage, eukaryotic cells activate a highly conserved intracellular signaling pathway known as the DNA damage response (DDR) [1,2]. This complex pathway allows cells to detect genomic damage and to mount an appropriate cellular response, from pausing the cell cycle and repairing DNA damage, to entering senescence, to undergoing apoptosis [1,2]. However, while many of the molecular components of the DDR are highly conserved across eukaryotic evolution, there is profound variation in how different cells respond to DNA damage based on such factors as signaling pathway status, tissue context, cell cycling status, development stage, and more [3–10]. Facing the same type and amount of DNA damage, cells in different contexts can vary widely in their propensity to undergo apoptosis in the face of DNA

**Funding:** This work was supported by the National Institutes of Health (5R24OD026435 to NP) and the Charles King Postdoctoral Fellowship (to BEC), and NP is an HHMI Investigator. No funder played any role in study design, data collection or analysis, decision to publish, or preparation of the manuscript.

**Competing interests:** The authors have declared that no competing interests exist.

**Abbreviations:** arm, *armadillo*; Ci, Cubitus interruptus; CRC, colorectal cancer; CRISPRa, CRISPR activation; Dcp1, Death Caspase-1; DDR, DNA damage response; DSB, double-strand break; EGFR, epidermal growth factor receptor; IR, irradiation; pERK, phosphorylated ERK; rho, *rhomboid*; RNAi, RNA interference; sgRNA, single guide RNA.

damage. For example, aberrant signaling pathway activity in many tumor types leads to a phenomenon known as radioresistance, in which tumor cells survive levels of DNA damage caused by radiation therapy that would induce apoptosis in similar nontumorous cells [11]. In contrast, some cell types are exquisitely sensitive to DNA damage. Human pluripotent stem cells, for example, have a remarkably low tolerance for DNA damage and undergo apoptosis in response to double-strand breaks (DSBs) caused by CRISPR/Cas9, which are routinely withstood by many other cell types [6]. In comparison with our understanding of the DDR pathway itself, much less is known about what drives differential sensitivity to genome damage in vivo.

Among the numerous cell signaling pathways implicated in modulating the DDR, the Wnt signaling pathway has been shown to interact with the DDR in a variety of contexts. The Wnt pathway is a highly conserved cell signaling pathway with critical functions in development, in adult stem cell populations, and in tissue homeostasis, and its dysregulation is implicated in a variety of diseases [12–14]. In humans, excess Wnt signaling correlates with radioresistance in a variety of tissue contexts and cancers (reviewed [15]). For example, in human colorectal cancer (CRC), activated Wnt signaling is considered the key driver of cancer progression, and functional studies have also demonstrated that Wnt signaling promotes radioresistance in CRC [15,16]. When human CRC cells are sorted based on levels of a Wnt reporter into Wnt[high] and Wnt[low] populations, Wnt[high] cells display significant resistance to radiation, and treatment with an inhibitor of the β-catenin/TCF interaction increases radiosensitivity [15]. In addition, treating nontumorigenic epithelial cell lines with Wnt pathway agonists leads to resistance to irradiation (IR) and to chemoradiotherapy [16].

A variety of mechanisms linking Wnt signaling to the DDR have been proposed in mammalian systems. One study in cultured CRC cells demonstrated the direct transcriptional activation of the critical DNA repair component *Lig4* by the transcription factor TCF downstream of Wnt signaling, in a process independent of p53 status [15]). Other studies have identified a converse phenomenon: down-regulation of Wnt signaling via p53 or E2F activity in response to DNA damage [17–22]. Altogether, it remains unclear whether these observed interactions between Wnt signaling and the DDR are generalizable or conserved in different contexts.

In the fruit fly *Drosophila*, signaling via the major Wnt ligand Wingless (encoded by the *wg* gene, the fly ortholog of *WNT1*) plays critical roles in growth and patterning, including in the larval precursor of the adult wing, the wing imaginal disc [23]. In addition, *wg* has been implicated in radioresistance in one particular context in this tissue, a region of the disc termed the "frown," which displays remarkable resistance to DNA damage [8,9]. The "frown" refers to a band of cells fated to become the hinge between the adult wing and notum, which can withstand high levels of IR without undergoing apoptosis, in a process that requires Wg signaling and JAK/STAT signaling, and which is mediated by regulation of the proapoptotic gene *reaper* [8,9]. These damage-resistant cells then contribute to the regeneration of the wing pouch following damage-induced apoptosis. In addition, a recent study using a specialized Gal4 system driven by the effector caspase Drice has shown that modulating Wnt signaling in the wing disc can affect both the apoptotic response to high levels of radiation and the ability for cells to survive low levels of caspase activation [24]. However, there remain open questions regarding the mechanisms connecting Wnt to the DDR and regarding whether Wnt signaling promotes resistance to DNA damage in other cellular contexts.

Here, using a CRISPR/Cas9-based approach to genetically inflict consistent levels of DNA damage in vivo, we demonstrate that loss-of-function of canonical Wnt signaling in the larval wing disc sensitizes these cells to DNA damage and biases them towards apoptosis. In contrast, Wg overexpression biases them away from apoptosis. We show that this function is mediated via expression of *rhomboid* (*rho*), which encodes a protease required for processing and

secretion of ligands of the epidermal growth factor receptor (EGFR) pathway, and that the effects of Wg loss-of-function can be rescued by activation of the EGFR pathway. This Wnt-mediated effect on the DDR requires the highly conserved components of the DDR *Chk2*, *p53*, and *E2F1*, and the proapoptotic factor *hid*. Altogether, we demonstrate that in the Wg signaling promotes cell survival in the face of DNA damage during development of the *Drosophila* wing.

## Results

### *wg* Loss-of-function sensitizes wing disc cells to DNA damage

During the course of a previous study of Wnt ligands in *Drosophila* [25], we made an unexpected observation implicating *wg* in the response to DNA damage in the developing wing. In that study, we used a collection of transgenic flies expressing single guide RNAs (sgRNAs) targeting each possible pairwise combination of the 7 *Drosophila* Wnt ligands (2 sgRNAs per target gene, 4 sgRNAs total per transgenic flies), to test for possible genetic interactions among these paralogous ligands. We used *hh-Gal4* to drive *UAS*:*Cas9.P2* (hereafter referred to as UAS:Cas9) and *UAS*:*sgRNA* constructs in the posterior of the developing wing disc and targeted each Wnt ligand both singly and in each pairwise combinations. For single knockouts (KOs), as expected, *wg* was the only Wnt ligand that displayed a loss-of-function phenotype in this tissue: loss of the posterior wing margin (**S1A Fig**).

Strikingly, when we performed double KOs of *wg* in combination with any of the other 6 Wnt ligands, we observed a dramatic phenotype: small, misshapen wings resembling those caused by massive cell death during development [26,27] (**S1B and S1D Fig**). In contrast, every pairwise double KO of Wnt ligands that did not include *wg* appeared morphologically wild type, indicating this small wing phenotype was not a generic response to the DNA DSBs caused by somatic CRISPR using 4 sgRNAs, but was instead specific to *wg* KO (**S1B and S1D Fig**).

To test whether this adult wing phenotype corresponds with increased apoptosis during development, we stained third instar larval (L3) wing discs from some these crosses with an antibody against cleaved Death Caspase-1 (Dcp1), a marker of apoptosis [28] (**S2 Fig**). Both *wg* single KO and *wg* + *wnt6* double KO wing discs displayed high levels of apoptosis in the posterior compartment, compared with various single KO and double KO wing discs (**S2 Fig**).

We reasoned that this small wing phenotype was unlikely to be the result of actual functional redundancy among the Wnt ligands because several of these ligands are not expressed in the developing wing pouch (*wnt5*, *wntD*, and *wnt10*) [25,29] and because the phenotype was similar in each case. We instead hypothesized that *wg* loss-of-function led these cells to have an increased sensitivity to the DNA damage caused by Cas9-induced DSBs. In other words, we proposed that whereas a wild-type wing disc can withstand the amount of DNA damage caused by somatic CRISPR and ultimately develop normally, a disc with compromised Wnt signaling is sensitized to DNA damage and undergoes far greater amounts of apoptosis.

To test this hypothesis, we generated 2 independent sgRNA constructs, each of which targets both *wg* and a random intergenic region (2 sgRNAs per target). When we coexpressed these constructs with *UAS*:*Cas9* using *hh-Gal4*, we observed the same small wing phenotype as above. This indicated that the wing phenotype we observed in all *wg* KO genotypes is not caused by redundant functions between Wg and other Wnt ligands but in fact can be phenocopied by targeting *wg* while simultaneously inducing DSBs at random intergenic loci (**S1C and S1D Fig**). As controls, we tested one of these intergenic sgRNAs either alone or in combination with either *wnt2* or *wnt10* and observed wild-type wing morphology in all cases (**S1C and S1D Fig**). Lastly, we used *hh-Gal4* to target *evi/wntless*, which is required for secretion of all Wnt genes except WntD [30], and observed the same well-characterized phenotype seen in

a *wg* loss-of-function wing (**S1C and S1D Fig**), further indicating that the small wing phenotype is not the result of genuine epistasis of multiple Wnt ligands, but is instead likely driven by increased apoptosis in discs with compromised Wnt signaling. However, given the mosaic nature of gene KO in somatic CRISPR [31,32], and given the fact that Wg is known to mediate cell competition in mosaic tissues [33], we wished to test this hypothesis using a more consistent and controlled means to manipulate Wg levels.

## A Cas9-based tool for genetically encoded DNA damage indicates that *wg* signaling alters the DDR in wing discs

To clarify and extend these observations, we designed a genetic system utilizing Cas9 to reliably generate moderate amounts of DNA damage, while separately manipulating Wnt signaling independently of CRISPR. We used *hh-Gal4, tubGal80$^{ts}$* to drive *UAS:Cas9*, together with either a nontargeting sgRNA or a random intergenic sgRNA (2 sgRNAs per construct), shifting to the Gal4-permissive temperature for 24 hours prior to dissection. We then measured apoptosis using cleaved Dcp1 antibody staining, coupled with Cubitus interruptus (Ci) as an anatomical marker of the anterior compartment. We quantified the volume of Dcp1+ cells in the posterior compartment relative to the anterior compartment, which served as an internal control in each disc. In the presence of a control RNA interference (RNAi) construct, we observed a modest but significant increase in apoptosis when the intergenic sgRNA was used compared to a nontargeting sgRNA, reflecting wild-type DDR activity in this assay (**Fig 1A and 1B**). When we reduced Wnt signaling using RNAi against either the *wg* ligand or the downstream effector *armadillo* (*arm*, the fly ortholog of β-catenin), we observed a significantly larger increase in apoptosis upon DNA damage (**Fig 1A and 1B**). Importantly, this increased apoptosis was not due simply to the reduction in Wnt signaling, as nontargeting sgRNA controls in these conditions did not lead to higher apoptosis (**Fig 1A and 1B**). When we overexpressed Wg using *UAS:wg*, we observed a suppression of apoptosis in the presence of CRISPR-mediated DNA damage (**Fig 1A and 1B**). Together, these results indicate that, for a given level of DNA damage, cells with reduced Wnt signaling are more likely to undergo apoptosis than wild-type cells, whereas cells with increased Wnt signaling are less likely to apoptose.

We confirmed these results using an independent Gal4 driver, *nub-Gal4*, which is expressed throughout the wing pouch (**S3A Fig**), and using an independent sgRNA targeting an additional gene that does not have an apoptotic phenotype, *yellow* (**S3C Fig**). We also performed this assay using an independent Cas9 transgene, *UAS:u$^{M}$Cas9*, which was designed to minimize the cell toxicity of Cas9 itself [34], and observed the same effect of Wnt signaling, but with lower levels of apoptosis in all conditions (**S3D Fig**). Importantly, while the u$^{M}$Cas9 construct does indeed cause lower levels of cell toxicity than more highly expressed Cas9 transgenes in the absence of any sgRNA (**S3E Fig**), we observed widespread apoptosis caused by CRISPR-based DSBs using any sgRNA we tested, including genes not expressed in the wing (**S3E Fig**), indicating that the Cas9-mediated process of cleaving DNA leads to substantial amounts of apoptosis in this tissue, independently of any toxic effects of Cas9 overexpression.

As an independent test of the effects of Wnt signaling on DNA damage, we examined the effects of 2 separate drugs that damage DNA, cisplatin and pirarubicin, as well as X-rays, in wild-type versus Wnt-compromised wing discs. Cisplatin and pirarubicin, the latter of which is a derivative of doxorubicin with lowered toxicity, are chemotherapy drugs that cause DNA damage [35,36]. In a pilot experiment, we confirmed that, similar to X-ray IR, the proapoptotic of these drugs requires *p53* and *Chk2* (known as *lok* in flies) [37]. In the absence of either drug, neither *wg-RNAi* nor *arm-RNAi* led to appreciable levels of apoptosis (**Fig 2A and 2B**), and control larvae fed with either drug for 24 hours prior to dissection exhibited apoptosis

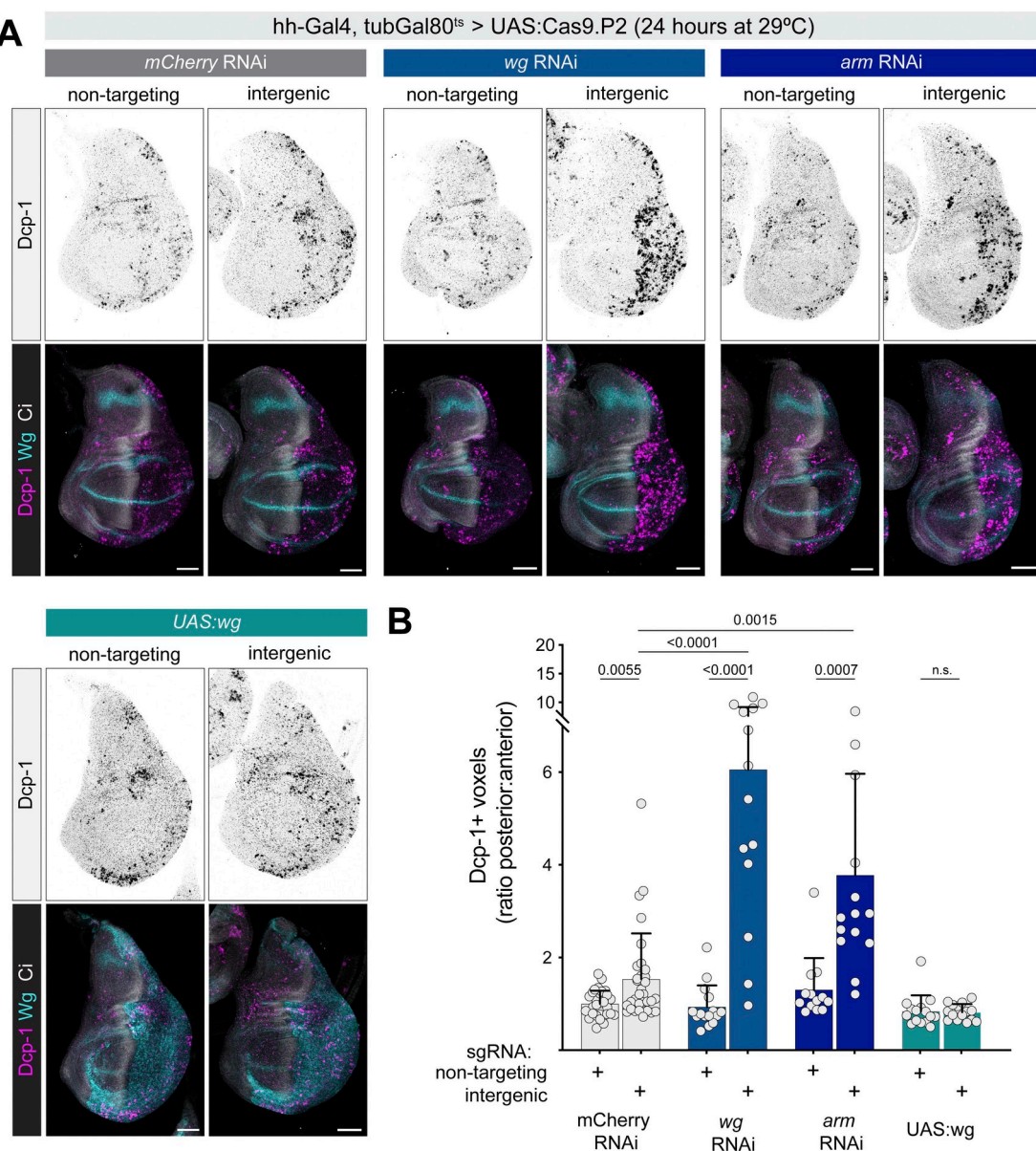

**Fig 1. Wg signaling buffers wing disc cells against DNA damage caused by CRISPR-induced DSBs.** (**A**) *hh-Gal4*, *tubGal80^{ts}* driving *UAS:Cas9.P2* and either a nontargeting sgRNA or an intergenic sgRNA, in the presence of other UAS:RNAi or overexpression constructs. Dcp1 antibody staining marks apoptotic cells, and Ci antibody marks the anterior of the wing disc, which serves as an internal control in each disc. (**B**) Quantification of apoptosis shown in (**A**). Dcp1+ voxels were quantified in a confocal stack and normalized to the *mCherry* RNAi + nontargeting sgRNA control. *P* values from a Student *t* test are shown, with Welch corrections for any comparisons with unequal variances. Scale bars = 50 μm. In this and all figures, dorsal is up and anterior is to the left. The data underlying the graphs shown in the figure can be found in S1 Data. Ci, Cubitus interruptus; Dcp1, Death Caspase-1; DSB, double-strand break; RNAi, RNA interference; sgRNA, single guide RNA.

throughout the wing disc. However, in *hh-Gal4 > wg-RNAi* or *hh-Gal4, tubGal80^{ts} > arm-RNAi* discs fed with either drug, we observed a significant enrichment of cell death specifically in the posterior (Gal4-positive) compartment (**Fig 2A and 2B**). Similarly, when we exposed such larvae to 1,000 RADs of X-rays, we observed a significant increase in apoptosis in the posterior of *wg-RNAi* and *arm-RNAi* compared to control discs (**S4 Fig**). In each of these

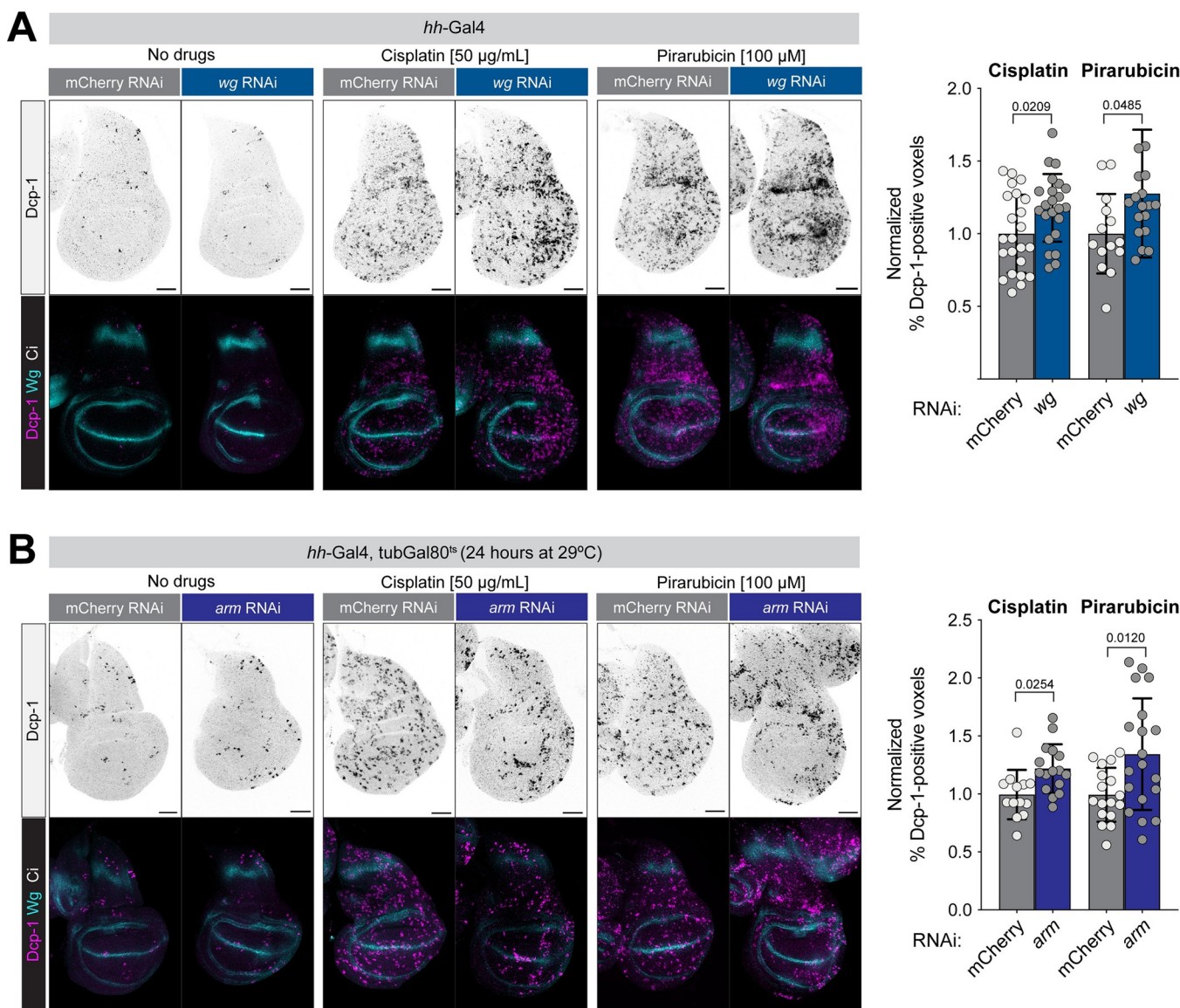

**Fig 2. RNAi against *wg* or *arm* sensitizes wing disc cells to the DNA-damaging drugs cisplatin and pirarubicin.** L3 larvae were fed cisplatin or pirarubicin for 24 hours and then assayed for apoptosis using an antibody against Dcp-1. (**A**) *wg* RNAi was constitutive throughout development, while (**B**) *arm* RNAi was restricted to the 24 hours before dissection to avoid tissue lethality. *P* values from a Student *t* test are shown, with Welch corrections for any comparisons with unequal variances. Scale bars = 50 μm. The data underlying the graphs shown in the figure can be found in S1 Data. *arm*, *armadillo*; Dcp1, Death Caspase-1; RNAi, RNA interference.

experiments, we used *tubGal80^{ts}* to limit *arm-RNAi* to a 24-hour period prior to dissection because *arm-RNAi* led to a near-total ablation of wing disc tissue when expressed throughout development. We note that the effect of *wg-RNAi* on X-ray sensitivity, while statistically significant, was relatively modest in absolute terms (**S4 Fig**), which may reflect differences in the response to acute X-ray exposure compared to a sustained 24-hour DNA damage induced using Cas9-mediated DSBs, and/or differences in the nature of the DNA damage caused by these agents.

To test the effect of Wg overexpression, we used CRISPR activation (CRISPRa) to transcriptionally activate the endogenous *wg* locus using *en-Gal4, tubGal80^{ts} > UAS:dCas9-VPR + sgRNA-wg*. We observed a significant reduction in apoptosis within the Wg-overexpression domain upon cisplatin, pirarubicin, or 1,000 RADs or X-ray damage (**S5 Fig**). Altogether,

these results suggest that Wg signaling in the wing disc promotes survival rather than apoptosis upon DSB damage. In each case, however, we note that apoptosis was not blocked altogether by Wg overexpression, indicating that the Wnt pathway is one of multiple factors that influences the DDR.

## Candidate suppressor screen places the canonical DDR downstream of Wnt-mediated DNA damage sensitivity

To characterize the mechanisms that link Wnt signaling to apoptosis upon DNA damage, we conducted a candidate suppressor screen focused on members of the DDR pathway and various signaling pathways that might act downstream of Wg. We utilized a single transgenic sgRNA construct targeting both *wg* and an intergenic region (*pCFD6-wg$^{2x}$-intergenic$^{2x}$*) that causes high levels of apoptosis in the wing disc when combined with *hh-Gal4, tubGal80$^{ts}$ > UAS:Cas9* after 24 hours at 29˚C (**Figs 3** and **S6**). We then combined this with a suite of loss- or gain-of-function reagents for various screen candidates and screened for suppressors that reduce the amount of apoptosis (**Figs 3** and **S6**).

Several core members of the DDR pathway were strong suppressors of Wnt-mediated DNA damage sensitivity. Previous studies have established that *p53*, *E2F1*, and *Chk2/lok* are essential for effectuating the DDR and apoptotic response to high levels of X-rays damage [37,38], whereas the highly conserved *Chk1* (*grps*), *ATR* (*mei-41*) are dispensable for this response [37] and *ATM* (*tefu*) has a more modest effect on the X-ray–induced DDR [39–42]. Consistent with these observations, we found that knockdown of *p53* using RNAi or a dominant negative allele, knockdown of *E2F1* using either RNAi or overexpression of *Rbf*, and RNAi against *Chk2/lok* ortholog strongly suppressed apoptosis in our screen (**Figs 3C, 3E** and **S6C**). In contrast, knockdown of *Chk1/grps*, *ATR/mei-41*, or *ATM/tefu* did not suppress apoptosis in our screen, consistent with their phenotypes in the context of X-ray damage. Studies in the wing disc and other tissues have shown that knocking down *cycA* causes endocyling by skipping M phase and that such endocycling cells are resistant to apoptosis [43–45]. Consistent with this, *cycA-RNAi* strongly suppressed apoptosis in our suppressor screen (**S6C Fig**). Altogether, our results suggest that the increased apoptosis we observe in Wg-compromised discs is mediated via the canonical DDR including *Chk2*, *p53*, and *E2F1*.

To test which of the 4 proapoptotic genes (*hid*, *rpr*, *skl*, and *grm*) are primarily responsible for mediating the apoptosis downstream of DNA damage in *wg* loss-of-function discs, we knocked down each using RNAi in a *hh-Gal4, tubGal80$^{ts}$ > UAS:Cas9, pCFD6-wg$^{2x}$-intergenic$^{2x}$* background and screened for a suppression of apoptosis. Only *hid-RNAi* suppressed apoptosis in this context (**Figs 4A, 4B** and **S7**). In contrast, 3 separate RNAi constructs targeting *rpr* failed to suppress this phenotype, as did RNAi against *grim* or *skl*. While these results suggest that *hid* is the primary effector of apoptosis in this context, we cannot rule out the possibility that the RNAi reagents failed to fully reduce the function of their target and that *rpr* or another proapoptotic gene may also contribute to this effect, especially in light of the fact that Wg is known to regulate in *rpr* in the wing disc in the context of IR [8]. Using a *hid* reporter, *hid-EGFP*, we observed that *hid* transcription increases significantly in the context of CRISPR targeting of *pCFD6-wg$^{2x}$-intergenic$^{2x}$* (**Fig 4C and 4D**). Together, these results suggest that *hid* is a key mediator of apoptosis downstream of *Chk2*, *p53*, and *E2F1* in a *wg* loss-of-function wing disc.

## Wg signaling acts via EGFR to dampen apoptosis upon DNA damage

We wished to know whether Wnt signaling acts directly on members of the DDR pathway, or whether it acts via a secondary signaling pathway. To identify candidate pathways that could mediate the effect of Wnt signaling, we performed a suppressor screen focused on a number of

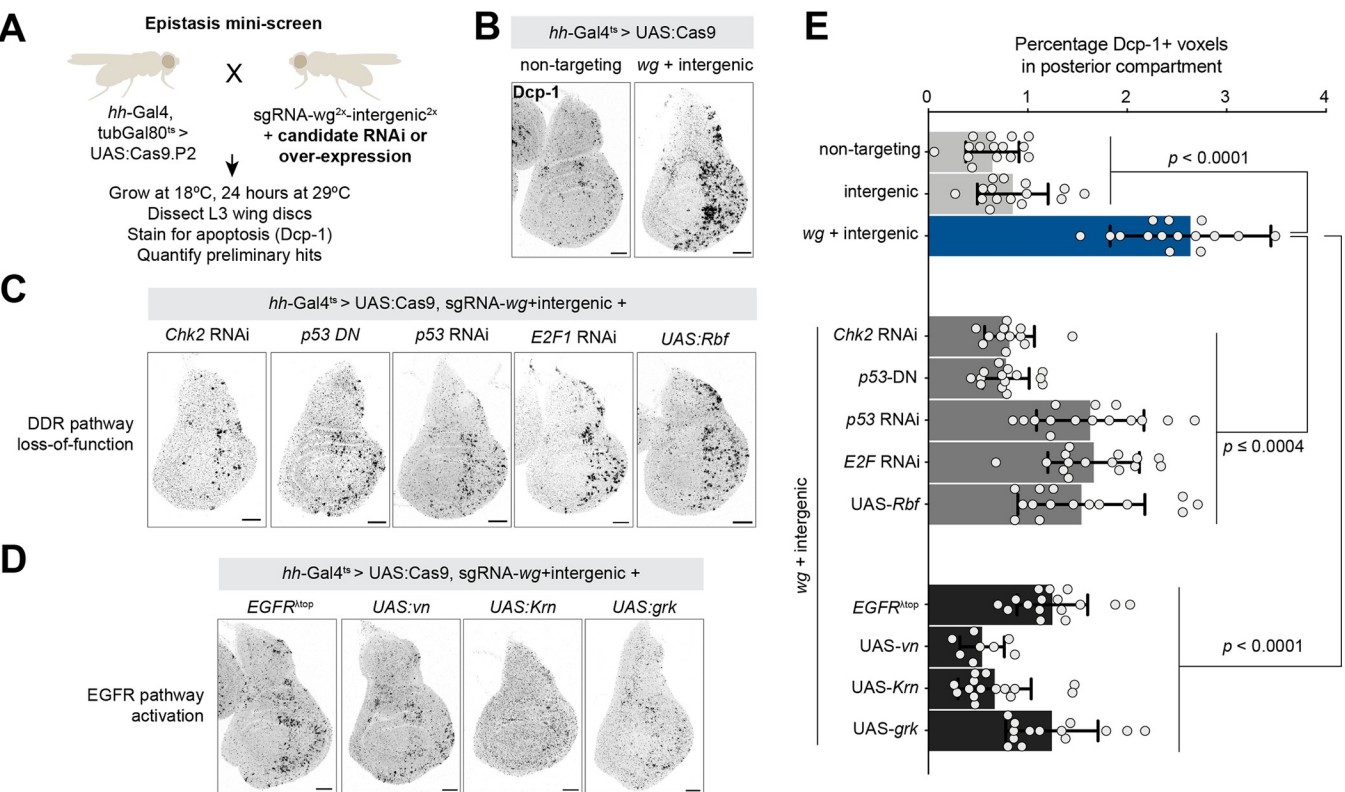

**Fig 3. Candidate screen identifies Chk2, p53, E2F, and the EGFR pathway as suppressors of *wg*-mediated DNA damage sensitivity. (See S6 Fig for complete primary screen results.)** (**A**) Screen schematic. (**B**) Representative wing discs showing background levels of apoptosis observed with a nontargeting sgRNA, and apoptosis levels in a double CRISPR targeting *wg* and an intergenic region. (**C**) *wg*-mediated DNA damage sensitivity is suppressed by loss-of-function reagents for Chk2, p53, or E2F, and by overexpression of UAS:Rbf, (**D**) and by gain-of-function reagents for the EGF pathway. (**E**) Quantification of results shown in (**B**-**D**). *P* values from a Student *t* test are shown, with Welch corrections for any comparisons with unequal variances. Scale bar = 50 μm. The data underlying the graphs shown in the figure can be found in S1 Data. EGFR, epidermal growth factor receptor; sgRNA, single guide RNA.

highly conserved signaling pathways: Hippo, EGFR, Hh, Dpp, JNK, JAK/STAT, Notch, as well as the transcription factor Myc, which is a known target of Wnt signaling [46]. As above, we either up-regulated or down-regulated components of each pathway in the presence of *hh-Gal4, tubGal80^{ts} > UAS:Cas9 + sgRNA-wg^{2x}-intergenic^{2x}* and tested whether these manipulations could suppress the effects of Wg KO on apoptosis levels upon DNA damage.

This candidate screen identified 2 putative hits that suppressed the apoptotic response to DNA damage in Wg-compromised discs: activated EGFR (*UAS:EGFR^{λtop}*) and activated Yki (*UAS:yki^{3SA}*) (**Figs 3** and **S6**). To test whether these pathways act downstream or in parallel to Wnt signaling, we examined the expression of pathway reporters in discs with altered Wg signaling. In *wg-RNAi* discs, the Hippo pathway reporter *ex-LacZ* [47,48] was unchanged (**S8 Fig**), indicating that Wg signaling does not modulate the Hippo pathway in this tissue and that the suppressive effect of activated Yki on apoptosis is likely a parallel process independent of Wnt signaling.

Separately, we noted that JNK signaling is known to play a role in wing disc regeneration following tissue damage [49] and that reduced JNK signaling dampens the apoptotic response to IR [50]. Consistent with this, we observed that the JNK pathway reporter *puc-lacZ* [51] was increased in Wnt-compromised DNA damage discs (**S8 Fig**). However, our suppressor screen demonstrated that blocking JNK signaling with a dominant negative form of *Bsk* [52] failed to suppress apoptosis in this context (**S6C Fig**). We confirmed this finding by overexpressing *puc*, a potent negative regulator of JNK, which also did not cause a significant reduction in

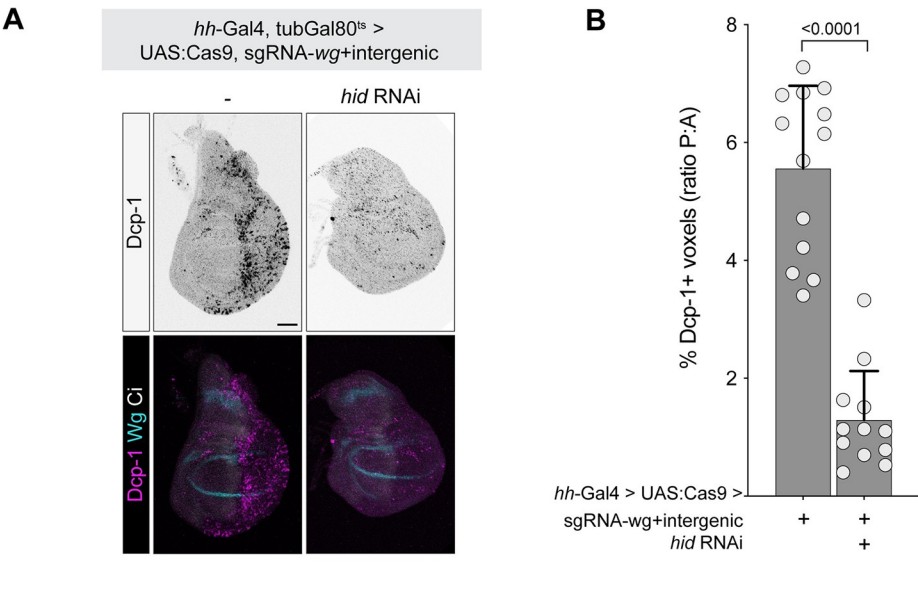

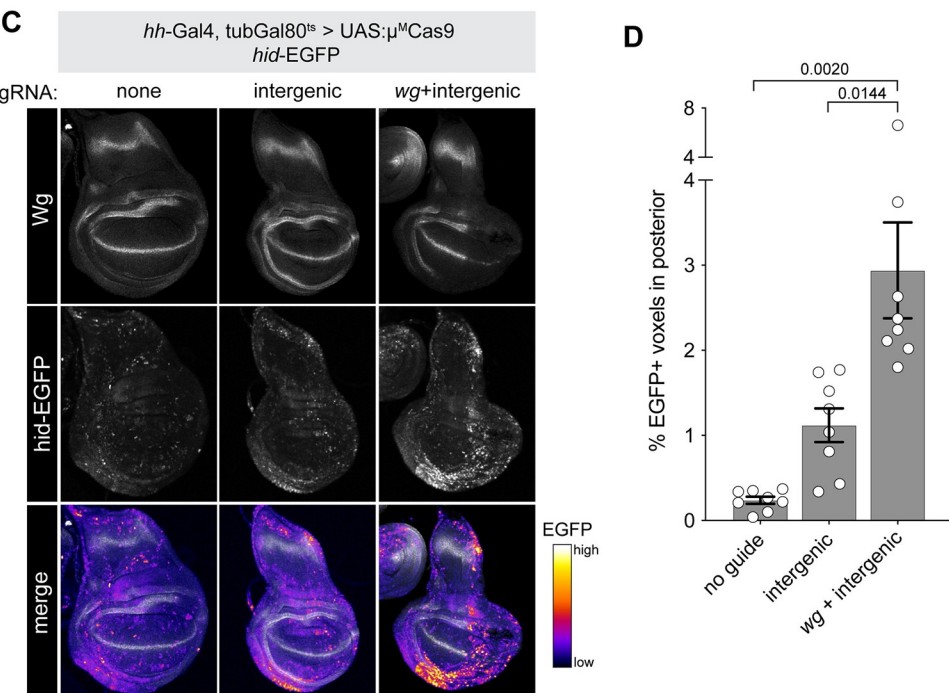

**Fig 4. *wg*-mediated DNA damage sensitivity acts via *hid*. See S5 Fig for related data.** (**A**) *hid* RNAi suppresses the apoptotic effect of double CRISPR targeting *wg* and an intergenic locus. (**B**) Quantification of data represented in (**A**). (**C**) A *hid-EGFP* reporter is activated by double CRISPR targeting *wg* and an intergenic locus. (**D**) Quantification of data shown in (**C**). *P* values from a Student *t* test are shown, with Welch corrections for any comparisons with unequal variances. Scale bar = 50 μm. The data underlying the graphs shown in the figure can be found in S1 Data.

apoptosis (S8 Fig). Thus, we conclude that JNK activation is downstream of cell death in this context, rather than a signaling component between Wnt signaling and the DDR.

EGFR signaling was an intriguing candidate in this context for a number of reasons. EGFR has been previously shown to oppose apoptosis by suppressing activity of the proapoptotic

gene *hid* at the transcriptional level and via phosphorylation [53,54], and ERK activation in the wing disc suppresses apoptosis in response to IR, via *hid* [55]. In addition, EGFR is known to act downstream of several patterning pathways in the early embryo to suppress cell death [56]. In eye imaginal discs, variable sensitivity to E2F-mediated apoptosis is driven by variation in EGFR signaling levels [57]. Importantly, there is also evidence that Wnt signaling directly regulates EGFR signaling in other contexts: in the developing leg imaginal disc, Wg signaling activates EGFR signaling via direct transcriptional activation of the EGFR ligand *vein* (*vn*) as well as the *rho* gene, which encodes a protease essential for processing and secretion of the EGFR ligand *spitz* [58]. These data suggest that EGFR could both be regulated by Wnt signaling and also be a potent suppressor of the apoptotic response to DNA damage.

We first confirmed that activated EGFR signaling suppresses the increased apoptosis found in DNA-damaged Wg-compromised disc by ectopically overexpressing several different EGFR ligands (Vn, Krn, Grk), all of which suppressed apoptosis in this context (**Fig 3E**). We then examined the expression of *vn* and *rho* transcripts via in situ hybridization, as well as phosphorylated ERK (pERK; *erk* is known as *rolled* in *Drosophila*—we hereafter refer to it as *erk* for simplicity) via antibody staining, in L3 wing discs with varying Wnt pathway manipulations. Using *hh-Gal4*, *tubGal80^{ts}* to drive either *wg-RNAi*, *arm-RNAi* in wing discs for 24 hours, we observed a dramatic reduction in *rho* transcription as well as pERK signal, while *UAS-wg* led to a striking up-regulation of both *rho* transcripts and pERK signal in the wing pouch (**Fig 5A**). We noted that this *rho* overexpression was limited to the wing pouch and did not extend to the notum (**Fig 5A**), suggesting that additional mechanisms likely mediate the effects of Wg signaling outside of the wing pouch. We did not observe notable changes in *vn* transcripts in any of these contexts, although we cannot rule out modest differences in expression levels in this assay (**S9 Fig**). Together, these results suggest that Wg signaling is both necessary and sufficient for *rho* transcription and pERK activation in the wing pouch.

To test whether the effect of Wg overexpression of pERK was indeed mediated via expanded *rho* expression, we used *hh-Gal4*, *tubGal80ts* to drive either *UAS-wg* alone or *UAS-wg + rho-RNAi* and measured pERK activity. In the presence of *rho-RNAi*, overexpression of Wg did not cause an increase in pERK signal (**Fig 5B and 5C**). This result demonstrates that Wg acts via Rho to activate ERK signaling in the wing disc.

To test whether *rho* and *erk* are functionally required for mediating the effect of Wg signaling on the DDR pathway, we tested whether overexpression of Rho could rescue the excess apoptosis phenotype. In the presence of *UAS:rho*, the effect of CRISPR targeting *wg+intergenic* was significantly ameliorated, suggesting that ectopically supplied Rho can indeed rescue the effect of reduced *wg* signaling (**Fig 6A and 6B**). We then tested whether knockdown of *rho* or *erk* would sensitize wing disc cells to DNA damage caused by targeting an intergenic region via CRISPR. Indeed, both *rho-RNAi* or *erk-RNAi* caused a significant increase in Dcp1-positive cells compared to control RNAi, specifically in the context of DNA damage but not in the presence of a nontargeting sgRNA, essentially phenocopying the effect *wg*-RNAi or *arm*-RNAi (**Fig 6C and 6D**).

Lastly, we tested whether specific knockdown of *rho* or *erk* would abolish the antiapoptotic effect of Wg overexpression. In these experiments, we expressed *UAS:wg*, which normally has the effect of reducing the apoptotic response to DNA damage (**Fig 6E**). However, when we knocked down either *rho* or *erk* using RNAi in the presence of *UAS-wg*, we observed a significant increase in the apoptotic response to DNA damage, indicating that antiapoptotic effect of *UAS:wg* requires both *rho* and *erk* (**Fig 6F**). Altogether, these data support a model in which Wg signaling acts via *rho* to activate EGFR signaling to oppose *hid* activation, likely at both the transcriptional and posttranslational level, in the context of DNA damage (**Fig 7**).

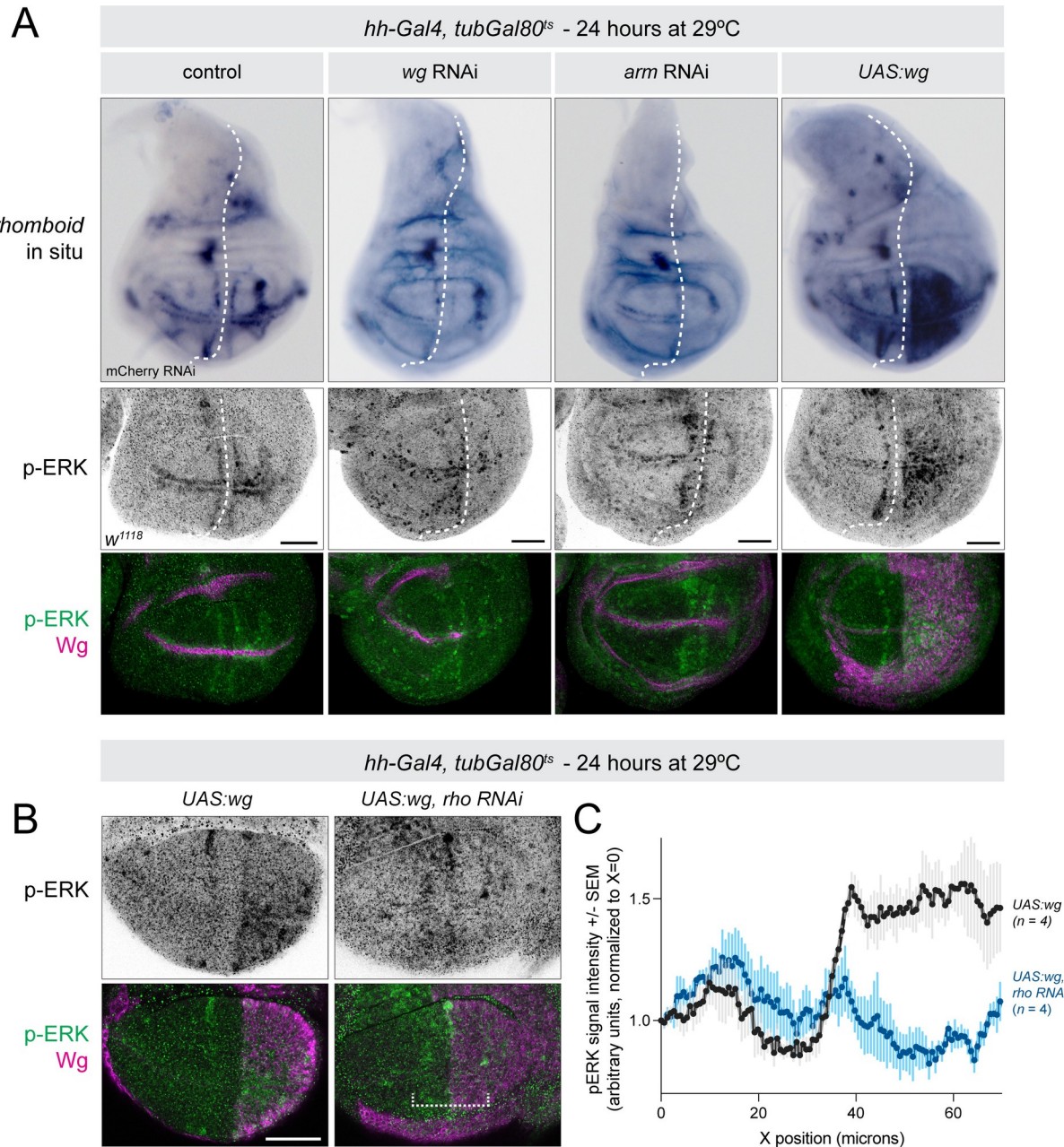

**Fig 5. *wg* signaling is necessary and sufficient for *rho* transcription and pERK expression in the wing pouch. (A)** Top row: in situ hybridization for *rho* in the indicated genotypes. Bottom two rows: antibody staining for pERK and Wg in the wing pouch. The dotted white line indicates the anterior–posterior boundary, with *hh-Gal4* expression restricted to the posterior. Both *rho* expression and pERK signal are reduced in the posterior in *wg* RNAi or *arm* RNAi discs and increased in the pouch (but not notum) following Wg overexpression. **(B)** Ectopic pERK expression in hh-Gal4, tubGal80ts > UAS-wg discs is abolished by rho-RNAi. The dotted white bracket indicates the width of the region of interest for quantification of p-ERK signal shown in **(C)**. Z-slices are shown for pERK antibody images. Scale bar = 50 μm. The data underlying the graphs shown in the figure can be found in S1 Data. *arm*, *armadillo*; pERK, phosphorylated ERK; *rho*, *rhomboid*; RNAi, RNA interference.

Several lines of evidence suggest that the protective effects of Wnt signaling on the DDR are relevant at moderate, but not very high, levels of DNA damage. While we have shown that *wg-RNAi* or *arm-RNAi* leads to excess cell death at moderate levels of DNA damage such as 1,000 RADs of X-ray IR, Cas9 targeting of an intergenic region, or moderate drug treatment, it has

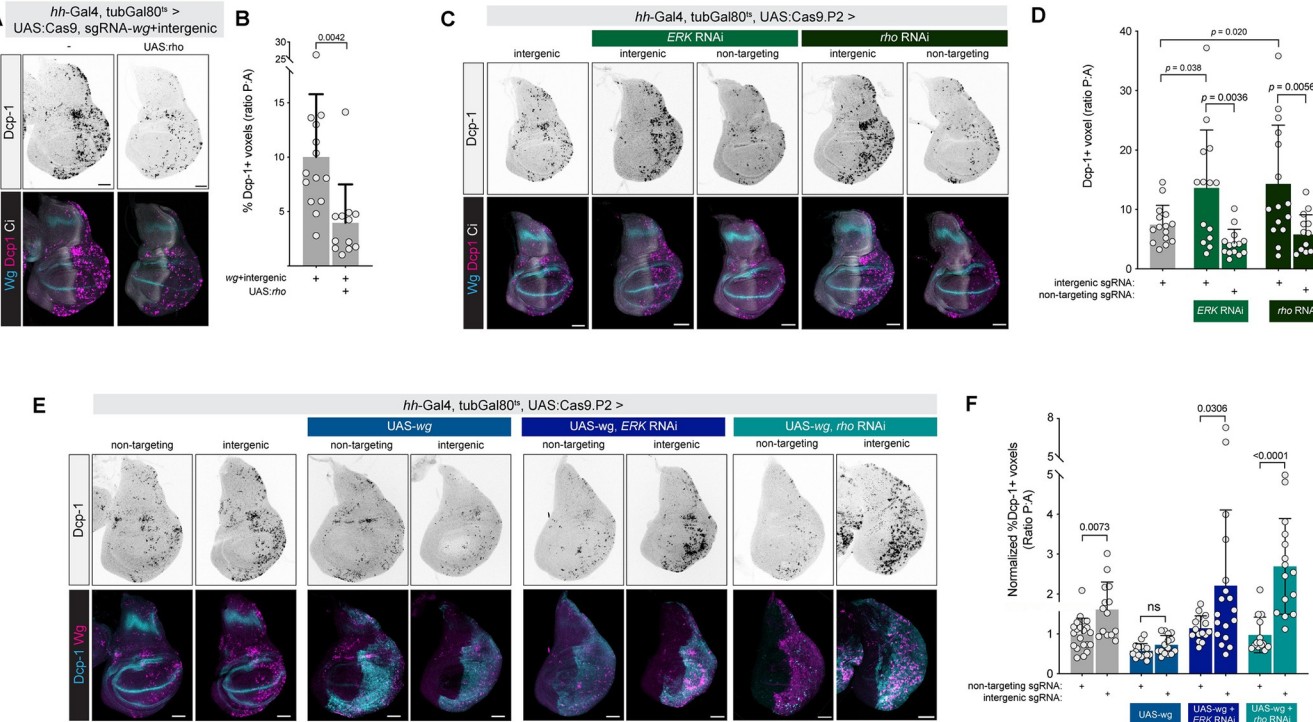

**Fig 6. Wg acts through Rho to modulate the DDR in the wing disc.** (**A**) Overexpression of *rho* using *UAS:rho* reduces apoptosis caused by double CRISPR against *wg* and an intergenic locus. (**B**) Quantification of data represented in (**A**). (**C**) RNAi against erk or *rho* sensitizes cells to CRISPR-induced DNA damage, phenocopying *wg* RNAi and *arm* RNAi (**D**) Quantification of data from (**C**). (**E**) *rho* and *erk* are each required for the apoptosis-suppressing effects of *wg* overexpression. *UAS-wg* alone suppresses apoptosis caused by CRISPR-induced DSBs. This suppressive effect is abolished by either *erk* RNAi or *rho* RNAi. (**F**) Quantification of data represented in (**E**). *P* values from a Student *t* test are shown, with Welch corrections for any comparisons with unequal variances. Scale bar = 50 μm. The data underlying the graphs shown in the figure can be found in S1 Data. *arm*, *armadillo*; DDR, DNA damage response; DSB, double-strand break; *rho*, *rhomboid*; RNAi, RNA interference.

been previously that high levels of X-ray damage (4,000 RADs) lead to cell death across the wing disc regardless of Wnt signaling pathway activity [8]. In addition, we observed that Wg overexpression cannot dampen the effects of very high levels of activation of p53 driven by a *UAS-p53* transgene (**S10 Fig**). Together, these suggest that the moderating effects of Wnt signaling on the DNA are overcome at high levels of DDR activation. Instead, we hypothesize that the biological function of Wnt in this context is to buffer wing development against moderate amounts of genotoxic damage. It remains unclear whether this role for Wnt signaling in tissues aside from the wing disc.

## Discussion

Wnt signaling is critical for the proper growth and patterning of the *Drosophila* wing disc, as well as regeneration following tissue damage [23,26]. Here, we characterize an additional role for this pathway during development: as a protective buffer against apoptosis in the context of DNA damage. We show that wing discs with reduced Wg signaling are sensitized to DNA DSBs and become biased towards apoptosis, whereas Wg overexpression leads to the opposite effect. This effect is mediated via the activity of core DDR effectors *Chk2*, *p53*, and *E2F1*, as loss-of-function of any of these factors abolishes the effect of Wnt signaling on the DDR and acts primarily via *hid*. We show that this effect of Wnt is upstream of EGFR signaling and is modulated via transcription of the ligand-processing protease *rho*. Wnt signaling is both necessary and sufficient for *rho* transcription in the wing pouch, and reducing either *rho* or the

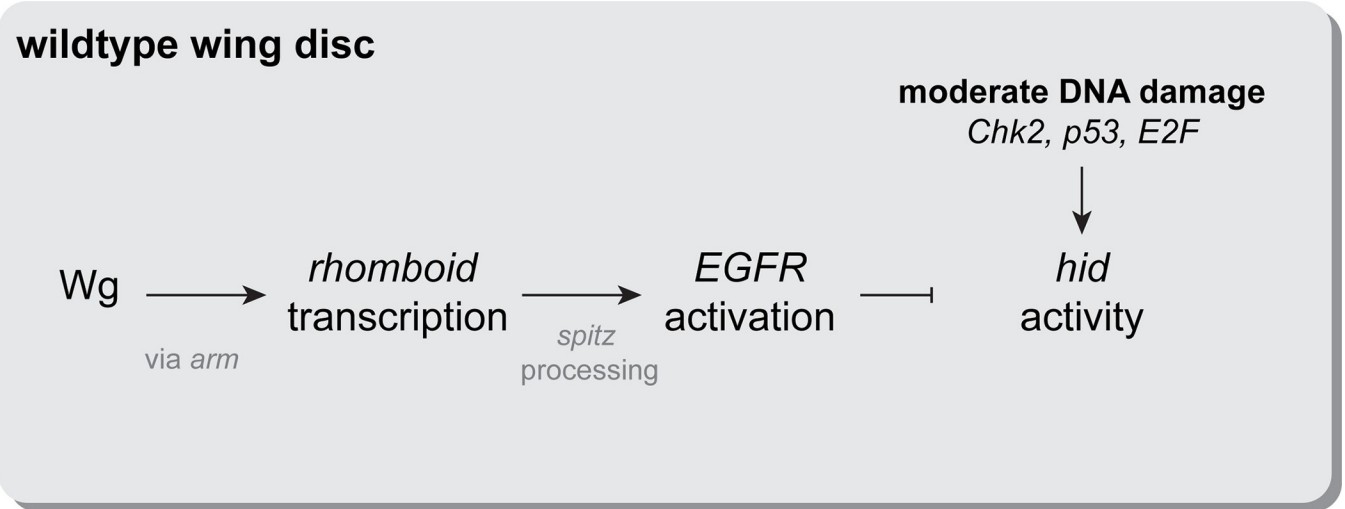

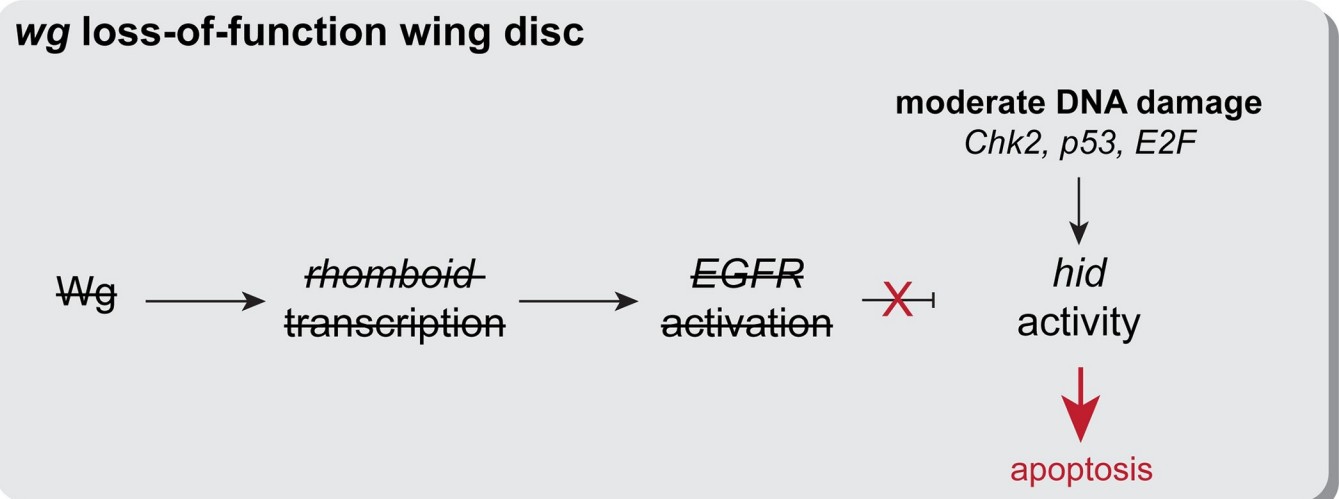

**Fig 7. Proposed model for the effect of Wnt signaling on the DDR pathway in the *Drosophila* wing.** During wild type development, *wg* acts via canonical signaling to activate *rho* transcription in the wing pouch. *rho* activity in turn leads to EGFR activation, likely through the processing of the *spitz* ligand, which acts in opposition to the DDR pathway in the context of moderate DNA damage. When *wg* signaling is compromised, the reduction in EGFR signaling biases cells towards apoptosis in the presence of moderate DNA damage. DDR, DNA damage response; EGFR, epidermal growth factor receptor; *rho*, rhomboid.

EGFR effector *erk* in the context of Wg overexpression abolishes the protective effect of Wnt signaling.

Previous studies have shown that Wg plays an important role in wing disc regeneration following injury, where Wg is up-regulated dramatically following tissue damage [26,27]. We note that in all of our experiments where we caused DNA DSBs using Cas9, DNA-damaging drugs, or X-rays, we did not detect any increased Wg signal via antibody staining (**Figs 1, 2, 6 and S2–S5**). Thus, we conclude that our observations represent a separate phenomenon, whereby Wnt signaling also plays a role in dampening apoptosis and tissue damage itself in the face of DSBs.

Our data suggest that the effects of Wnt on the DNA damage pathway operate upstream of EGFR signaling. EGFR signaling has been demonstrated to play a critical prosurvival role in a wide variety of contexts in flies and many other organisms. For example, in the early *Drosophila* embryo, multiple signaling pathways act via EGFR signaling to promote cell survival, in the

absence of DNA damage [56]. Our data suggest that impinging on EGFR signaling may be a common feature of pro-growth, antiapoptotic signaling pathways.

Our findings rely on experimental manipulation of Wg and EGFR signaling levels in the wing disc, but we note that during wild type development, cells at different positions within the wing disc naturally experience varying levels of both Wg and EGFR signaling. This could imply that, absent any experimental manipulations, cells could vary in their sensitivity to DNA damage. Indeed, some previous studies have shown that apoptosis in response to X-rays tends to concentrate in areas of low EGFR signaling [7], and others have shown that some, but not all, areas of high Wnt signaling—specifically, the "frown" located between the dorsal edge of the wing pouch and the notum—are resistant to X-ray IR [8,9]. However, in our experiments, outside of the damage-resistant "frown," we did not observe any consistent spatial patterns of apoptosis in wild type discs following damage with CRISPR-induced DSBs, X-rays, or the DNA-damaging drugs cisplatin and pirarubicin. We suggest that the relatively low levels of Wg signaling that are experienced by the entire disc [59] are sufficient to confer the moderate resistance to DNA damage that we describe here.

Wnt signaling has been linked to radioresistance in a variety of human cancers, and previous studies in human CRC cell lines have suggested that this effect is mediated through the regulation of Lig4, a ligase central to the DDR [15]. Here, we provide evidence that, in the context of *Drosophila* development, Wnt signaling acts via EGFR signaling to promote resistance to DNA damage. Further study is needed to ascertain whether this mechanism operates in human cancers as well. In addition, given the complex relationship between Wnt signaling and cell cycle control [60], which is itself intimately related to the DDR, we believe that future studies of the relationship between Wnt signaling, cell cycle, and the DDR will be valuable.

## Materials and methods

### Experimental animals

*Drosophila melanogaster* lines used in this study are listed in S1 Table (previously described lines) and S2 Table (sgRNA lines), and genotypes are provided for each figure in S3 Table. Crosses were maintained on standard cornmeal fly food, except as indicated for drug treatments, at maintained at either 18°C, 25°C, or 29°C as indicated in the text.

### Antibody staining and confocal imaging

Third instar larval wing discs were dissected in PBS, fixed for 25 to 30 minutes in 4% paraformaldehyde in PBS, stained using standard immunohistochemistry protocols, counterstained with DAPI (1:1,000) and mounted in Vectashield mounting medium (Vector Labs) for confocal imaging. The following antibodies were used in this study: rabbit anti-Dcp1 (Cell Signaling Technologies Cat. 9578, 1:100), mouse anti-Wg (Developmental Studies Hybridoma Bank 4D4, 1:100), rat anti-Ci (Developmental Studies Hybridoma Bank 2A1, 1:10), rabbit anti-phospho-ERK (Phospho-p44/42 MAPK, Cell Signaling Technologies 4370S, 1:500), and rabbit anti-GFP AlexaFluor488 conjugate (Molecular Probes, 1:300). Alexa Fluor 488, 555, and 647 coupled secondary antibodies were used at a concentration of 1:400. Wing discs were imaged using either a Zeiss LSM 780, LSM 980, or an Olympus IX83 confocal microscope, through the Microscopy Resources of the North Quad (MicRoN) facility at Harvard Medical School.

### Quantification of apoptosis via Dcp1 staining

Apoptosis was quantified as the percentage of voxels (the three-dimensional equivalent of pixels), which stained positive for Dcp1 antibody in a confocal z-stack. By measuring the

percentage rather than the absolute area of Dcp1+ cells in the posterior compartment, this measurement accounts for variation in the volume of the compartment. Confocal z-stacks were analyzed in FIJI by manually selecting 2 separate region of interests for the anterior and posterior disc compartments based on either Ci staining (which marks the anterior compartment) or morphological landmarks, then using the "Voxel Counter" plug-in (https://imagej.net/ij/plugins/voxel-counter.html) to quantify the percentage of Dcp1-positive voxels. As an internal control to account for inter-experiment variability in background signal, we calculated the ratio of Dcp1+ voxels in the Gal4-positive posterior compartment to the Gal4-negative anterior compartment.

When calculating the posterior:anterior ratio, to account for the fact that the percentage of Dcp1+ positive cells in the control compartment (the denominator of in our ratio calculation) was often close to zero and therefore could be sensitive to very small variations, we thresholded each image in such a way to introduce low, uniform levels of nonspecific noise or "speckling" across the tissue. To ensure that our findings are robust across different methods of Dcp1+ quantification, we compared 3 different methods for the data presented in our Fig 1: the posterior:anterior ratio of Dcp1+ voxels (ultimately presented above), the absolute percentage of Dcp1+ voxels in the posterior compartment (not normalized to the anterior, to avoid dividing by a small number), and the posterior:anterior ratio after adding a value of 1.0 to every measurement, to shift all values away from zero and thus reduce variation. All significant differences between treatments were robust across these 3 methods.

Given the number of samples required for the suppressor screen, we only measured the percentage of Dcp1-positive voxels in the posterior compartment, not normalized to the anterior, which we observed to give highly concordant results and allowed us to process a far larger number of samples. All graphs were created and statistical tests performed using Prism (GraphPad).

### Quantification of pERK signal following Wg overexpression

Confocal z-slices of pERK-stained wing discs were analyzed using an approximately 70-μm rectangular region of interest centered on the anterior–posterior boundary and located in the ventral wing pouch. pERK signal was quantified in FIJI using the "Plot Profile" feature. Values were normalized to the anterior-most value (x = 0) for each sample.

### Adult wing scoring and imaging

Adult wings were mounted on a glass slide in a 1:1 mixture of Permount and xylenes and imaged on a Zeiss Axioskop 2 using brightfield optics. Wings were categorized into phenotypic 4 categories as shown in S1 Fig.

### Drug treatment

Cisplatin (ApexBio A8321) and pirarubicin (Selleck Chemicals S1393) were diluted to the indicated concentrations in distilled water, which was then used to rehydrate Formula 4–24 Instant Blue Food (Carolina Biological Supply.) Larvae were placed on drug food approximately 24 hours prior to dissection at the L3/wandering stage. Pilot studies at a range of dilutions identified the 50 μg/mL cisplatin and 100 μM pirarubicin as concentrations sufficient to cause widespread apoptosis across the wing disc without killing the animal. Doxorubicin was also tested in these pilot experiments but caused larval lethality at concentrations below those necessary to cause widespread apoptosis in the wing disc.

## X-ray treatment

Flies of the appropriate genotype laid in a standard fly bottle for an overnight egg collection. At the L1 stage, 50 to 60 larvae were transferred to individual fly vials. At the third instar larval stage, experimental vials were subjected to 1,000 RADs in a TORREX 120D X-ray Inspection System (ScanRay Corporation) and dissected 4 hours later for antibody staining. Pilot experiments at 100, 500, 1,000, and 2,000 RADs identified this exposure level displayed a differential sensitivity to X-rays in *wg* RNAi discs.

## In situ hybridization

In situ hybridization experiments were performed as described in [25]. Antisense probes against *rho* were synthesized using primers F: ggccgcggGTCAGTTGCGTGCGAGC R: cccggg gcGCATAGACGCCACCGCT and against *vein* using F: ggccgcggAATAAAAACAACAACA GTGCAACA and R: cccggggcATTTCCGTTTATCCTGCAAATACT. These primers contain overhangs (shown in lowercase), which allow for the addition of a T7 site in a second PCR.

## Supporting information

**S1 Table. Sources and genotypes of Drosophila lines used in this study.**
(DOCX)

**S2 Table. Drosophila sgRNA lines used in this study.**
(DOCX)

**S3 Table. Genotype table.**
(DOCX)

**S1 Fig. (Related to Fig 1). CRISPR KO of wg sensitizes developing wing tissue to DNA damage.** (**A**) Somatic single CRISPR KOs of each Wnt ligand in the posterior of the developing wing. Single KO of *wg* produces a loss of the wing margin in the posterior, whereas no other Wnt ligand displays a phenotype. (**B**) Double CRISPR KOs of each pairwise comparison of Wnt ligands using *hh-Gal4*. In combination with any other Wnt ligand, *wg* causes a dramatic defect in wing development, indicative of excessive cell death. All other pairwise combinations appear wild type. (**C**) Double CRISPR KO of *wg* with 2 separate intergenic sgRNA sequences causes severe wing defects, whereas double KO of an intergenic sequence with *wnt2* or *wnt10* produces no phenotype. The phenotype of *wntless* single KO is reminiscent of *wg* KO alone. (**D**) Scoring of wing defects shown in (**A**-**C**). Posterior is down in all wing images. The data underlying the graphs shown in the figure can be found in S1 Data.
(DOCX)

**S2 Fig. (Related to Fig 1) Apoptosis in CRISPR KO wing discs.** Wing discs of the indicated genotypes stained for Dcp1 to visualize apoptotic cells. The adult phenotypes shown in S1 Fig correspond to increased apoptosis in the posterior of the wing discs.
(DOCX)

**S3 Fig. (Related to Fig 1). Additional validation that wg signaling modulates the response to DNA damage caused by somatic CRISPR in the wing disc.** (**A**) As an alternative to *hh-Gal4*, *nub-Gal4* driving UAS:Cas9.P2 throughout the wing pouch sensitizes cells to DNA damage caused by CRISPR targeting of an intergenic region. (**B**) *wg* RNAi in the disc posterior sensitizes wing disc cells to DNA damage caused by CRISPR targeting of an intergenic region, and (**C**) against a sgRNA targeting the *yellow gene*, which has no apoptotic phenotype by itself. (**D**) *hh-Gal4* driving a lower-toxicity variant of Cas9, u^MCas9, also sensitizes wing disc cells to

DNA damage caused by CRISPR. (**E**) Somatic CRISPR in the wing disc using a lower-toxicity variant of Cas9, u^MCas9, causes substantial apoptosis with a wide variety of sgRNAs targeting intergenic sequences, genes expressed in the wing disc, and genes not expressed in the wing disc (*osk*). *P* values are shown from Student *t* test, with Welch correction for any comparison with unequal variances. The data underlying the graphs shown in the figure can be found in S1 Data.
(DOCX)

**S4 Fig. (Related to Fig 2.) RNAi against wg or arm sensitizes wing discs to 1,000 RADs of X-ray damage.** *hh-Gal4* or *hh-Gal4, tubGal80^ts* was used to drive RNAi against *wg* or *arm*, respectively, and larvae were subjected to 1,000 RADs of X-rays 4 hours prior to dissection. The amount of apoptosis was quantified by measuring the percentage of Dcp1+ voxels in the posterior (Gal4 on) versus anterior (Gal4 off). In the case of *wg* RNAi, the effect of *wg* RNAi on apoptosis levels appeared more pronounced in the notum rather than the wing pouch. The data underlying the graphs shown in the figure can be found in S1 Data.
(DOCX)

**S5 Fig. (Related to Fig 2). wg overexpression using CRISPRa dampens apoptosis caused by X-rays and DNA-damaging drugs.** *wg* was overexpressed in the posterior wing disc using *en-Gal4, tubGal80^ts > UAS:dCas9-VPR*, and flies were subjected to DNA damage caused by (**A**) 1,000 RADs of X-rays 4 hours prior to dissection, (**B**) pirarubicin for 24 hours, or (**C**) cisplatin for 24 hours. Dotted lines represent the approximate boundary of the posterior compartment in control discs (identified via UAS:GFP expression) or the regions where excess Wg is detected via antibody staining in CRISPRa tissues. Scale bars are 50 μm, posterior is the right, and dorsal is up. Wg signal is displayed using the "Fire" lookup table in FIJI/ImageJ. *P* values are shown from Student *t* test, with Welch correction for any comparison with unequal variances. The data underlying the graphs shown in the figure can be found in S1 Data.
(DOCX)

**S6 Fig. (Related to Fig 3). Candidate suppressor screen identifies members of the DDR pathway, the Hippo pathway, and the EGFR pathway as suppressors of DNA damage-induced apoptosis in the presence of compromised Wnt signaling.** (**A**) Schematic of the candidate suppressor screen. *hh-Gal4, tubGal80^ts > UAS:Cas9.P2* + pCFD6-*wg-intergenic* was used to drive DNA damage and apoptosis in the posterior wing disc, in the presence of various UAS-driven RNAi or other functional transgenes targeting the DNA damage repair pathway and various signaling pathways. Discs were stained for Dcp1 and a primary screen qualitatively identified major changes in the amount of apoptosis in the wing disc. (**B**) Control discs show the levels of Dcp1 signal seen in representative discs with a nontargeting sgRNA (negative control) and with sgRNAs targeting *wg* and an intergenic region (positive control.) Primary screen hits are shown in pink for members of the (**C**) DNA damage pathway and (**D**) various signaling pathways. These hits were secondarily screened and quantified as shown in Fig 3.
(DOCX)

**S7 Fig. (Related to Fig 4). Candidate suppressor screen identifies hid as the effector of apoptosis in the context of DNA damage in Wnt-compromised discs.** The same screening format as in S6 Fig was used to screen RNAi lines targeting the DIAP1 inhibits rpr, hid, skl, and grm. Of these constructs, 2 hid RNAi lines suppressed apoptosis in this context.
(DOCX)

**S8 Fig. (Related to Fig 3) The effects of Wnt signaling on the DDR pathway are not mediated via Hippo or JNK signaling.** (**A**) *wg* RNAi in the posterior wing compartment does not

alter the expression of the Hippo signaling reporter *ex-LacZ*. (**B**) The JNK signaling reporter *puc*:*lacZ* is activated by DNA damage and apoptosis in the wing disc. However, as indicated in S6 Fig, suppressing JNK signaling does not suppress the apoptosis caused by DNA damage in a Wnt-compromised disc. The data underlying the graphs shown in the figure can be found in S1 Data.
(DOCX)

**S9 Fig. (Related to Fig 5) vein levels are not strongly modulated by varying wg signaling levels.** In situ hybridization against *vn* in the indicated genotypes.
(DOCX)

**S10 Fig. Wg overexpression does not suppress the apoptotic effect of high levels of p53 overexpression.** *UAS:p53* overexpression in the wing disc posterior causes massive apoptosis and tissue death within 24 hours (top row). This effect is not ameliorated by the coexpression of *UAS-wg* (bottom row).
(DOCX)

**S1 Data. Individual numerical values underlying all figures.**
(XLSX)

## Acknowledgments

We thank Tanuj Thakkar for assistance with in situ hybridizations, Rich Binari for assistance with fly work, and members of the Perrimon lab for valuable feedback. N.P. is an HHMI investigator. This article is subject to HHMI's Open Access to Publications policy. HHMI lab heads have previously granted a nonexclusive CC BY 4.0 license to the public and a sublicensable license to HHMI in their research articles. Pursuant to those licenses, the author-accepted manuscript of this article can be made freely available under a CC BY 4.0 license immediately upon publication.

## Author Contributions

**Conceptualization:** Ben Ewen-Campen, Norbert Perrimon.

**Data curation:** Ben Ewen-Campen.

**Formal analysis:** Ben Ewen-Campen.

**Funding acquisition:** Norbert Perrimon.

**Investigation:** Ben Ewen-Campen.

**Methodology:** Ben Ewen-Campen.

**Project administration:** Ben Ewen-Campen.

**Supervision:** Norbert Perrimon.

**Writing – original draft:** Ben Ewen-Campen, Norbert Perrimon.

**Writing – review & editing:** Ben Ewen-Campen, Norbert Perrimon.

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
