## [Editor Report · Decision Letter 0]

7 Feb 2024

Dear Norbert, 

Thank you for submitting your manuscript entitled "Wnt signaling modulates the response to DNA damage in the Drosophila wing imaginal disc by regulating the EGFR pathway" for consideration as a Research Article by PLOS Biology.

Your manuscript has now been evaluated by the PLOS Biology editorial staff as well as by an academic editor with relevant expertise and I am writing to let you know that we would like to send your submission out for external peer review.

Once your full submission is complete, your paper will undergo a series of checks in preparation for peer review. After your manuscript has passed the checks it will be sent out for review. To provide the metadata for your submission, please Login to Editorial Manager (https://www.editorialmanager.com/pbiology) within two working days, i.e. by Feb 09 2024 11:59PM.

Kind regards,

Ines

--

Ines Alvarez-Garcia, PhD

Senior Editor

PLOS Biology

---

## [Decision Letter · Decision Letter 1]

1 Mar 2024

Dear Norbert,

Thank you for your patience while your manuscript entitled "Wnt signaling modulates the response to DNA damage in the Drosophila wing imaginal disc by regulating the EGFR pathway" was peer-reviewed at PLOS Biology. It has now been evaluated by the PLOS Biology editors, an Academic Editor with relevant expertise, and by two independent reviewers. 

The reviews are attached below. As you will see, the reviewers find the conclusions interesting, but they also raise several issues that should be addressed before we can consider the manuscript for publication. Reviewer 1 thinks the ratio of Dcp1 signal in the posterior and anterior compartments should be normalised with another method to avoid potential errors. This reviewer also asks to consider alternative explanations to some of the findings and to improve the presentation. Reviewer 2 thinks it should be shown whether the increase in ERK levels caused by Wg overexpression is rescued by Rhomboid depletion to confirm that Wg signalling regulates ERK via Rho, and also asks for several clarifications and minor corrections.

In light of the reviews, we would like to invite you to revise the work to thoroughly address the reviewers' reports. Given the revisions needed, we cannot make a decision about publication until we have seen the revised manuscript and your response to the reviewers' comments. Your revised manuscript is likely to be sent for further evaluation by all or a subset of the reviewers.

**IMPORTANT - SUBMITTING YOUR REVISION**

3. Resubmission Checklist

a) *PLOS Data Policy*

b) *Published Peer Review*

d) *Blurb*

Please also provide a blurb which (if accepted) will be included in our weekly and monthly Electronic Table of Contents, sent out to readers of PLOS Biology, and may be used to promote your article in social media. The blurb should be about 30-40 words long and is subject to editorial changes. It should, without exaggeration, entice people to read your manuscript. It should not be redundant with the title and should not contain acronyms or abbreviations. For examples, view our author guidelines: https://journals.plos.org/plosbiology/s/revising-your-manuscript#loc-blurb

Sincerely,

Ines

--

Ines Alvarez-Garcia, PhD

Senior Editor

PLOS Biology

Reviewers' comments

Rev. 1:

Ewen-Campen and Norbert Perrimon report that Wingless (Wnt1) signaling in Drosophila larval wing discs represses apoptosis in cells with DNA damage by inhibiting the expression of pro-apoptotic Hid. Although Wg was shown before to inhibit apoptosis by repressing the expression of pro-apoptotic Reaper in cells with DNA damage, this is the first report to link it to Hid. Moreover, the authors find that regulation occurs via transcriptional activation of rhomboid and EGFR signaling. This mode of regulation operates at intermediate levels of DNA damage induced by Cas9 and in response to DNA damaging chemicals (the data for 1000R of X-rays is less convincing for reasons given below). The identification of a new regulatory crosstalk between conserved Wg and EGFR signaling pathways in controlling apoptosis in response to damaged DNA is significant and should interest the readers of PLoS Biology. The data support most of the conclusions but there are some concerns that need to be addressed. I believe this could be done without additional experiments.

Many of the graphs show the ratio of Dcp1 signal in P and A compartments. Because cell death is targeted to the P compartment in these experiments, the signal in the A compartment is close to zero. Dividing by a number that is close to zero can induce huge errors. Unexpected changes in the control A compartment can also add error. For example, in Fig. 2B, drug-induced Dcp1 signal in the A compartments appears lower in Arm RNAi discs than in mCherry RNAi discs, which could artificially bump up the P/A ratio. In another example, in Fig. S4, X-ray-induced Dcp1 signal in the A compartment is lower in Arm RNAi disc than in mCherry RNAi control. Differences in compartment size induced by genetic manipulations could also add error because Dcp1+ voxels will be less where there are fewer cells to begin with even if the same % of cells die. For these reasons, the authors should back up their key conclusions with a second method to normalize the signal, for example, by normalizing Dcp1+ area/volume in the P compartment to total P compartment area/volume.

Related to the above, it is hard to see how images shown in some figures could produce the quantified results in the graphs. For example, in Fig. 5S, Dcp1 signal in the A and P compartments do not seem different in the discs shown (which I assume are representative), yet quantification shows nearly 2-fold differences.

Hid is clearly involved but the negative data with rpr, skl and grim RNAi could be because respective RNAi constructs are not knocking down their targets sufficiently. Can this possibility be ruled out, especially for rpr which was shown previously to be transcriptionally regulated by Wg (Ref.8)?

Suggestions to improve the presentation:

The sentence that 'The cyclin protein cycA is known to be required for cell cycle arrest upon DNA damage [43], and as expected cycA-RNAi strongly suppressed apoptosis in our suppressor

screen (Figure S6C.)' does not make sense because Chk1 and ATR are also required for cell cycle arrest upon DNA damage and their RNAi had no effect. A more likely explanation is that cyclin A RNAi arrests cells at a stage in the cell cycle (because cyclin A is needed for normal cell cycle progression unlike Chk1 and ATR) where they are less likely to undergo apoptosis or where Cas9 works less efficiently.

Ref. 8-9 are cited for the role of Wg in inhibiting radiation-induced apoptosis in the hinge region of the wing disc but 'the downstream mechanisms connecting Wnt to the DDR remain unclear, as does the question of whether Wnt signaling promotes resistance to DNA damage in other cellular contexts'. This is inaccurate because Ref. 8-9 identified transcriptional repression of pro-apoptotic gene Rpr as the mechanism by which Wg inhibits apoptosis. The authors should acknowledge this previous finding. Furthermore, a newer paper from this lab reports that Wg signaling inhibits caspase activation in regions of the wing disc outside the radioresistant hinge (PMID: 38182576). Therefore, the authors may want to update their statement about other cellular contexts.

In Fig. S3E, do all panels also have Wg RNAi or Wg sgRNA? Otherwise, there are several like intergenic, yellow, and ebony that are inducing apoptosis on its own, which is contrary to the rest of the paper.

Please double check the references to figure numbers. For example:

Figure 3 legend refers to Figure S3, but it should be S6.

In the sentence 'Only hid-RNAi suppressed apoptosis in this context, which we confirmed with two independent RNAi constructs (Figure 5A, B and Figure S8.)', 5A, B should be 4A, B.

Rev. 2: Marco Milán - note that this reviewer has signed his review

The manuscript of Ewen-Campen and Perrimon unravels a role of the Wingless signalling pathway in protecting epithelial cells to DNA damage-induced apoptosis through induction of the EGFR signalling pathway. The ms is well written, topic is timely, figures self-explanatory and the conclusions are well based on very-well designed experiments. Everything started from the initial observation that specific depletion of the Wingless ligand in the Drosophila wing epithelium enhanced CRISPR/Cas9-driven DNA damage induced cell death when targeting other Wnt ligands. From there, authors demonstrate that Wingless signalling (through the canonical pathway) dampens the apoptotic effects of DNA damage produced by several means (CRISPR/Cas9-cleavage of intergenic regions, X-rays, chemical drugs). Interestingly, Wingless appears to be also sufficient to dampen cell death when overexpressed. Through a candidate approach, authors identify elements of the DDR and EGFR pathways and present evidence that Wingless signalling reduces cell death through transcriptional induction of rhomboid (known to cleave EGFR secreted ligands) and ERK activation which will lead to the blockage of hid expression/activity. Overall, I support publication of this ms once authors have dealt with the following issues:

(1) Does Wingless signalling have any effect on DNA-damage induced cell cycle arrest?

(2) The results of CycA on DNA damage induced cell death bother me as I see no connection with the rescue of cell death.

(3) Authors propose that Wg signalling regulates ERK through Rhomboid. In order to reinforce the proposed linear relationship between these elements, if would be necessary to show whether the increase in ERK levels caused by Wingless overexpression are rescued by rhomboid depletion.

(4) Labels are not correct in many figures. Ci stainings are lacking in many cases, the color of Dcp1 and Wg channels are interchanged, etc. Why the use of Ci antibody should be stated.

(5) I would show hid-EGFP (Figure 4C) in a single panel in B/W

(6) Bergmann and Steller demonstrated (a few years ago) that ERK blocks Hid protein directly through phosphorylation. Perhaps, this should be included in the model and discussed in the paper.

(7) Murcia et al 2019 presented evidence that DNA-damage induced ERK signaling dampens the apoptotic effects by repressing Hid. Perhaps, these data should be discussed in the paper.

(8) The no-effects of Bsk-DN on apoptosis worry me a lot. Have authors tested other UAS-transgenes able to block JNK more efficiently (eg. UAS-puc). Apparently, JNK is indeed involved in DNA damage induced cell death, if I recall correctly from the published literature.

(9) Please, include the complete genotypes in Table S1.

---

## [Editor Report · Decision Letter 2]

21 May 2024

Dear Norbert,

Thank you for your patience while we considered your revised manuscript entitled "Wnt signaling modulates the response to DNA damage in the Drosophila wing imaginal disc by regulating the EGFR pathway" for publication as a Research Article at PLOS Biology. This revised version of your manuscript has been evaluated by the PLOS Biology editors and the Academic Editor.

Based on on our Academic Editor's assessment of your revision, we are likely to accept this manuscript for publication, provided you satisfactorily address the data and other policy-related requests stated below.

We expect to receive your revised manuscript within two weeks. 

*Published Peer Review History*

*Press*

Sincerely,

Ines

--

Ines Alvarez-Garcia, PhD

Senior Editor

PLOS Biology

Fig. 1B; Fig. 2A, B; Fig. 3E; Fig. 4B, D; Fig. 5C; Fig 6B, D, F; Fig. S1D; Fig. S3A-D; Fig. S4; Fig. S5A-C and Fig. S8D

---

## [Editor Report · Decision Letter 3]

26 Jun 2024

Dear Dr Perrimon,

Thank you for the submission of your revised Research Article entitled "Wnt signaling modulates the response to DNA damage in the Drosophila wing imaginal disc by regulating the EGFR pathway" for publication in PLOS Biology. On behalf of my colleagues and the Academic Editor, Nicolas Tapon, I am delighted to let you know that we can in principle accept your manuscript for publication, provided you address any remaining formatting and reporting issues. These will be detailed in an email you should receive within 2-3 business days from our colleagues in the journal operations team; no action is required from you until then. Please note that we will not be able to formally accept your manuscript and schedule it for publication until you have completed any requested changes.

PRESS

Best regards,

Ines

--

Ines Alvarez-Garcia, PhD

Senior Editor

PLOS Biology
